# Unravelling the Roles of Bacterial Nanomachines Bistability in Pathogens’ Life Cycle

**DOI:** 10.3390/microorganisms12091930

**Published:** 2024-09-23

**Authors:** Romain Gory, Nicolas Personnic, Didier Blaha

**Affiliations:** Group Persistence and Single-Cell Dynamics of Respiratory Pathogens, CIRI—Centre International de Recherche en Infectiologie, CNRS, INSERM, Ecole Normale Supérieure de Lyon, Université Claude Bernard Lyon 1, 50 avenue Tony Garnier, 69007 Lyon, France; romain.gory@etu.univ-lyon1.fr (R.G.); nicolas.personnic@cnrs.fr (N.P.)

**Keywords:** nanomachines, type III secretion system, flagellum, phenotypic heterogeneity, community behavior, bet-hedging, division of labor

## Abstract

Bacterial nanomachines represent remarkable feats of evolutionary engineering, showcasing intricate molecular mechanisms that enable bacteria to perform a diverse array of functions essential to persist, thrive, and evolve within ecological and pathological niches. Injectosomes and bacterial flagella represent two categories of bacterial nanomachines that have been particularly well studied both at the molecular and functional levels. Among the diverse functionalities of these nanomachines, bistability emerges as a fascinating phenomenon, underscoring their dynamic and complex regulation as well as their contribution to shaping the bacterial community behavior during the infection process. In this review, we examine two closely related bacterial nanomachines, the type 3 secretion system, and the flagellum, to explore how the bistability of molecular-scale devices shapes the bacterial eco-pathological life cycle.

## 1. Introduction

Until the 2000s, it was believed that the behavior of microbial communities was driven by the genetic diversity found in multispecies consortia, where each bacterial species tends to specialize in performing particular tasks. In this view, the interaction between genetically diverse species was thought to be key to community function and stability [1]. Yet, conditions supporting a robust clonal bacterial expansion cause a rapid drop in genetic diversity. This is especially evident during the invasion of sterile tissue by microbial pathogens. The bacterial specialization then occurs primarily as the result of phenotypic heterogeneity, i.e., the reversible expression of alternative phenotypes in a part of a clonal population to improve the performance of the entire population [2,3]. Studying bacterial phenotypic variants is challenging, as it is a reversible and dynamic physiological state occurring only in a fraction of the bacterial population. Fortunately, advances in quantitative single-cell techniques have facilitated the exploration of this microbial diversity [4,5,6]. Bistability is one expression of bacterial phenotypic heterogeneity. It refers to the ability of a function to coexist in two distinct stable states under the same set of conditions [7,8]. Bistability also implies the capacity to toggle between active (“ON”) and inactive (“OFF”) states. Within an isogenic bacterial population, bistability can affect a multitude of processes, such as metabolic pathways altering the utilization of resources [9] and signaling pathways impacting cellular communication [10]. One major bacterial process involved in phenotypic heterogeneity and subject to bistability is nanomachines [11]. Bacterial nanomachines are complex molecular structures and systems that enable various essential functions. Some examples of these machineries include pili, hair-like appendages protruding from the bacterial surface, facilitate adhesion to surfaces, including host cells, and mediate DNA transfer during conjugation [12]. Other examples include secretion systems that are specialized complexes responsible for transporting small molecules, proteins, or DNA across bacterial membranes, enabling the delivery of toxins and virulence factors to extracellular space or target cells [13]. Last example include replisome machinery that ensures faithful duplication of the bacterial genome during cell division [14]. Among all these nanomachines, the injectisome and the flagellum are particularly noteworthy due to their key roles in bacterial pathogenesis [15]. The injectisome also referred to as the type III secretion system (T3SS) or molecular syringe, is found in Gram-negative bacteria. It allows bacterial effector proteins to be administered into eukaryotic host cells in order to bypass defense systems and hijack cellular functions [16,17]. The flagellum is an extracellular biological filament driven by a fast-rotating internal motor that propels bacteria into the environment. This nanomachine is of great important in both Gram-positive and Gram-negative bacteria and has multiple functions such as motility, chemotaxis, adhesion, and biofilm formation [18]. T3SS and flagellum are closely related in terms of evolution, and both play pivotal roles in triggering immune detection responses by the host, as their presence acts as a signal for the immune system to recognize and respond to bacterial invasion [19,20]. Both rely, at their core, on a homologous type III secretion apparatus for extracellular component export and, ultimately, needle or flagellar filament assembly. Bistable expression or activation of the injectisome was reported for *Dickeya dadantii*, *Escherichia coli*, *Pseudomonas aeruginosa*, and *P. syringae*, *Salmonella enterica* [21,22,23,24,25]. Furthermore, the flagellum exhibits bistability in more species, with, for instance, *Bacillus subtilis*, *Caulobacter crescentus*, *Clostridium difficile*, *Escherichia coli*, *Legionella pneumophila*, *Pseudomonas aeruginosa*, *Salmonella enterica,* and *Shewanella* spp. [26,27,28,29,30,31]. In both cases, many are known human pathogens. 

In this review, we introduce the notion of phenotypic heterogeneity before exploring the structure, regulation, and functions of the injectisome and flagellum. Finally, we discuss bistability in the expression of these nanomachines and the several questions that are raised: (i) the regulatory mechanisms involved, (ii) the impact on the bacterial populations, and (iii) its role during the infection process and treatment outcome.

## 2. Molecular Pathways Underlying Phenotypic Heterogeneity: Stochasticity versus Determinism

To gain a deeper understanding of the mechanisms driving the bistability of the two most renowned nanomachines, we provide an overview of the mechanisms that contribute to phenotypic heterogeneity. Variations among individuals arise from a multitude of molecular mechanisms, which can be broadly categorized into two main types: stochasticity and determinism. Stochasticity introduces randomness and unpredictability into bacterial evolution, which is crucial for adaptation and survival in fluctuating environments. In contrast, determinism results in phenotypic variation driven by specific, identifiable factors or rules, leading to predictable outcomes in the evolutionary trajectories of bacterial populations.

### 2.1. Stochasticity in Gene Expression

#### 2.1.1. Expression Noise

Expression noise refers to the variability or stochastic fluctuations in the levels of gene expression among individual cells within a population, even when those cells are genetically identical and in the same environment. This variability can arise due to several factors. Within bacterial cells, certain genetic and cellular functions are inherently more prone to noise compared to others. The activity of bacterial promoters, genes encoding metabolic enzymes, stress response pathways, quorum sensing and Cell–Cell communication, toxin-antitoxin systems, and flagellar biosynthesis and motility are known to be noisier [32,33,34,35]. Pioneering observation of variation in gene expression in *Escherichia coli* was revealed by differences in the expression of fluorescent reporters regulated by the same promoter, both within and between cells (Figure 1a) [36]. Such so-called noise influences the genetic circuit’s activity and, therefore, bacterial functionalities at the single-cell level within bacterial populations [27,37,38,39]. These fluctuations depend on intrinsic heterogeneity in enzymatic reactions, notably impacting transcriptional and translational rates, accentuated by the burst character of the transcription [39,40,41,42]. Irregular expression rates, coupled with positive feedback loops, can facilitate the emergence of distinct phenotypes within populations, enhancing physiological stability and serving as toggle switches [43]. This feedback amplification phenomenon is frequently essential to underlying mechanisms of phenotypic heterogeneity [8,43]. 

#### 2.1.2. Epigenetics

Epigenetics is an increasingly studied regulatory mechanism that can cause phenotypic diversity through alteration of DNA methylation patterns or nucleoid-associated proteins (NAPs) such as histone-like nucleoid-structuring proteins (H-NS). It can also contribute to the stochastic aspect of phenotypic heterogeneity [8,44,45,46]. For instance, stochastic mechanisms may involve random fluctuations in DNA methylation patterns, leading to sporadic expression of certain genes. In contrast, deterministic mechanisms could include the regulated binding of NAPs to specific genomic regions, influencing the expression of nearby genes in a predictable manner. Variations in these could modify chromosome conformation and accessibility, resulting in changes in gene expression, thereby creating heterogeneity, including bacterial persistence (Figure 1b).

#### 2.1.3. Asymmetric Partitioning

Cell division can contribute to stochastic phenotypic heterogeneity (Figure 1c). During cell division, the segregation of cytoplasmic molecules is probabilistic, resulting in variations between siblings [47]. Low-abundant compounds, such as enzymes and transcription factors, are the most likely to be unequally distributed. This may lead to differences in metabolic functions and gene expression [8]. Additionally, subcellular components, such as the inclusion body, can cause slower cell growth [9].

### 2.2. Determinism in Phenotypic Variation

#### 2.2.1. Intrinsic Cellular Factors

Even though stochastic variations can lead to phenotypic variation, this phenomenon can also be attributed to determinism. Cell division asymmetry is a source of microbial phenotypic heterogeneity. This process can lead to phenotypic variation in microorganisms, in which a few phenotypic traits depend on replicative aging. Such mechanisms have been well-demonstrated in yeasts [48,49] and in a few bacteria, such as *Mycobacterium*, where senescence leads to variable growth and susceptibility to antibiotics [50,51]. Additionally, without the concept of senescence, compounds in bacteria can be dynamically polarized, resulting in phenotypic variation in chemotaxis activities, sporulation, and signaling transduction proteins [52].

A second mechanism is periodic oscillations. These clock-based mechanisms regulate the phenotype switch in the population in response to cellular state or environmental signals. However, a shift in this clock between cells could cause a delay in switching, resulting in population heterogeneity. One of the best-understood mechanisms of this kind is the cell cycle of *Caulobacter crescentus*. Gene expression and protein synthesis vary in a cell cycle-dependent manner, coordinating the swarmer to stalked transition and cell divisions [53,54]. Another oscillator process was found in *Synechococcus elongatus*, a cyanobacterium with an internal circadian clock regulating a 24-h cyclic gene expression rhythm influenced by light (Figure 1d) [55]. 

#### 2.2.2. Cell–Cell Interactions

Cell–cell interactions represent another significant mechanism contributing to the development of phenotypic diversity in bacterial populations (Figure 1e). Recent research has unveiled novel physical interactions between *Myxococcus xanthus* and its prey [56], leading to the emergence of a prey-killing phenotype through the assembly of a type IV filament-like machinery. Similarly, filamentous cyanobacteria like *Anabaena* and *Nostoc* exhibit contact-dependent heterogeneity, inducing differentiation into azote-fixing heterocysts through shared periplasmic communication [57]. Quorum sensing, a well-studied form of cell–cell communication, involves diffusible molecules and is often perceived as a collective behavior. However, evidence suggests heterogeneity in the expression of quorum sensing-related genes [10,58,59]. Recent studies have revealed that quorum sensing and autoinducer molecules can also be drivers or targets of phenotypic heterogeneity. Phenotypic heterogeneity allows bacteria to optimally cope with consecutive, rapid, and frequent environmental fluctuations (bet-hedging) or to interact with each other by performing distinct, often complementary, functions (division of labor) [60]. Additionally, emerging studies have identified various other cell–cell interactions. Phenotypic heterogeneity in bacterial stress responses can arise from short-range cell-to-cell interactions, leading to a collective phenotype that protects a significant portion of the population. In *E. coli*, a study showed that this heterogeneity relies on cell-to-cell interactions, where cells protect each other from H_2_O_2_ through their individual stress responses [61]. Finally, among *Bacilli*, a widely distributed family of Rap (Response-regulator aspartyl phosphate) phosphatases is known to modulate the function of key regulators of phenotypic heterogeneity by controlling their phosphorylation. Additionally, their associated extracellular Phr (Phosphatase regulator) peptides act as signals, creating a cell-to-cell communication network that regulates the phenotypic development of the entire population [62].

#### 2.2.3. Heterogeneous Microenvironment

Lastly, phenotypic variation may also emerge due to micro-environmental heterogeneity (Figure 1f). Despite the small scale of a bacterial population, environmental conditions vary, and neighboring bacterial cells can be exposed to various concentrations of stimuli. For example, distinct subpopulations of *Yersinia pseudotuberculosis*, specialized or not in the expression of a nitric oxide detoxifying gene, were observed on the same infection site due to a nitrite oxide gradient released by the host in response to the infection [63]. For the marine decomposer *Vibrio splendidus*, resource gradient also leads to the emergence of specialized compartmented subpopulations. One non-motile, nitrogen utilizing, at the colony periphery, and a motile, carbon-storing core [64]. The same principle applies to biofilm, in which microscale chemical heterogeneity can explain most of the biological variations within populations [65,66].

### 2.3. Approaches and Challenges in Studying Phenotypic Heterogeneity

Studying phenotypic heterogeneity is challenging because phenotypic variants are a reversible phenomenon, representing only a few individuals out of thousands, and are often nested within complex environments. The study of phenotypic heterogeneity requires real-time observation methods due to its dynamic nature. These methods can be technically demanding and resource-intensive. For a long time, the most common approach was to manipulate bacterial populations, either genetically or environmentally, and measure the impact on phenotypic heterogeneity levels. This approach allowed for the identification of stresses and pathways involved in modulating phenotypic heterogeneity levels in a population. Significant breakthroughs have been made in recent years with the development of methods for tracking, collecting, and analyzing the phenotypic variants [67]. Now, the focus is on single-cell research using microfluidic devices combined with microscopy and flow cytometry. This permits the isolated individual bacteria to track their behavior over time using time-lapse microscopy, allowing researchers to examine the history of phenotypic variants [5,67].

Fluorescent reporters coupled with acquisition systems, such as automatic microscopy, have enabled these analyses. This approach has facilitated the generation of molecular maps of phenotypic variants, with a particular focus on persisters, known as antibiotic-tolerant cells. Bistability is also assessed using fluorescent reporters, such as pfla::GFP in *Legionella pneumophila*, which enables the observation of flagellum bistability associated with phenotypic heterogeneity [30]. The TIMER system distinguishes rapidly proliferating cells from slower ones by transitioning from green to red fluorescence [68]. Mechanistic insights can be gained through Fluorescent-Activated Cell Sorting (FACS) and omics methods, which facilitate subpopulation analysis like persisters at the molecular level [6].

## 3. The Injectisome Nanomachine

### 3.1. Injectisome Structure and Functions

Gram-negative bacteria have developed diverse nanomachines known as secretion systems to transport proteins, small molecules, and DNA to extracellular spaces or target cells [13]. Among these, the injectisome also referred to as the type III secretion system (T3SS), has been extensively studied. Its structure is well-documented and comprises several key components [15,16,69,70,71]. The injectisome comprises a basal body firmly anchored to the bacterial membrane. It is structured with circular rings, encompassing the outer membrane (OM) ring, the inner membrane (IM) ring, and the cytosolic C ring (Figure 2a). These rings, assembled via sec-dependent secretion, provide stability to the nanomachine [16]. At the base of the injectisome, there is a type III secretion apparatus responsible for secreting extracellular parts. The needle of the injectisome is linked to the base via an inner rod and comprises numerous secreted units of a monomeric protein arranged in a stacked configuration, with the tip unit being added last. Upon encountering a host cell, the injectisome assembles a translocon pore in the host cell membrane. Once the entire complex is formed, the injectisome switches to secreting effector proteins, which are delivered through the needle into the host cells [16].

The injectisome, or type III secretion system (T3SS), plays a pivotal role in mediating interactions between bacteria and eukaryotic hosts. Its primary function involves translocating virulence effectors directly into eukaryotic cells, making it a critical virulence factor for many pathogenic bacteria [16,17]. Variations in the function of T3SS are observed across bacterial species, exemplified by the *Salmonella* pathogenicity island 1 (SPI-1) T3SS in *Salmonella enterica*, which aids in the invasion of host cells [72], and the SPI-2 T3SS manipulating vacuole traffic for intravacuolar survival [73,74]. In *Shigella* spp., the T3SS enables escape from vacuoles, cytoplasmic survival, and modulation of host immunity [75]. Additionally, T3SS-mediated processes including modulation of host actin cytoskeleton and immune responses during extracellular infections [16,17]. These orchestrated mechanisms typically enhance the survival of bacterial populations and their ability to colonize the host. Interestingly, the injectisome also plays a role in symbiotic relationships, as observed in rhizobia, where effector translocation into plant cells promotes nodulation and symbiosis establishment [76].

### 3.2. Injectisome Regulatory Pathway

The regulatory network of the injectisome varies significantly across bacterial species, involving horizontal gene transfer [77], unlike the vertically conserved flagellum. In numerous bacteria, genes encoding Type III Secretion Systems (T3SS) are organized in operons, located either on the bacterial chromosome within pathogenic islands, as observed in *Salmonella enterica* [78], or on plasmids, as seen in *Shigella* spp. [79,80]. However, in *Chlamydia*, T3SS genes are distributed across the genome without significant differences in G+C content, suggesting no horizontal acquisition [77]. In *Pseudomonas aeruginosa*, the T3SS is encoded by multiple operons regulated by the major activator ExsA, which is antagonized by ExsD and PtrA (Figure 2b) [81,82]. Transcriptional control involves promoter sequences upstream of each operon, facilitating the coordinated expression through RNA-polymerase recruitment. Environmental signals such as low Ca^2+^ levels activate transcription by increasing cAMP concentration, mediated by proteins ExsC, ExsE, and ExsD. Conversely, high Ca^2+^ conditions or stressors like Cu^2+^, osmotic changes, or metabolic stresses repress T3SS transcription, highlighting the intricate regulatory mechanisms governing injectisome activity [81,82].

Regulation pathways governing injectisome gene expression vary between bacteria and can involve diverse environmental or cellular state signals. In *Salmonella*, for instance, the central transcriptional regulator HilA controls the SPI-1, which comprises injectisome genes. HilA’s regulation involves a positive feedback loop with three proteins responding to various environmental stimuli. Factors such as nutrient availability, including high sugar concentrations, inhibit SPI-1 transcription. In contrast, conditions like contact with bile, high Mg^2+^, and elevated fatty acid concentrations in the intestine promote its expression through a regulatory cascade [72].

### 3.3. Molecular Bases of the Bacterial Injectisome Bistability

Bistability of the injectisome has only been observed in a few pathogens, such as Salmonella and Pseudomonas. The most documented injectisome bistability phenomenon has been described in *Salmonella enterica*, in which the injectisome is encoded in SPI-1 with effector proteins [72]. This genomic region exhibits bistable expression [11,22,83], likely influenced by mechanisms involving Rho-dependent transcription termination and nucleoid structuring protein H-NS [22]. Inhibiting Rho-dependent transcription termination leads to the activation of H-NS-deactivated genes. Transcriptional noise from pseudo antisense promoters, when randomly not halted by Rho, may extend into a regulatory region associated with H-NS. This results in DNA decompression, rendering regulatory sequences accessible to the main SPI-1 regulator HilD and RNA polymerase, thereby activating SPI-1 through a positive feedback loop triggered by HilD [22]. Furthermore, flagellar genes in *Salmonella enterica* also display bistability and interact with SPI-1 genes [83]. Nutrients play a crucial role in governing the proportion of cells expressing flagellar genes [84], but this regulatory phenomenon has not been observed for the T3SS [83]. However, recent research has shown that acetate, in combination with nutrients, synergistically activates SPI-1 gene expression and enhances T3SS bistability through a positive feedback loop [83]. 

In *Pseudomonas aeruginosa*, despite the well-established regulation of the T3SS, the mechanisms underlying its bistability remain unclear. Nevertheless, it appears to be regulated through a cascade involving the cyclic-AMP-Vfr system, which acts as a global regulator of virulence gene expression, including the T3SS [23]. Mutants lacking Vfr show reduced expression of the T3SS, suggesting its role in bistability regulation [23]. Vfr regulates the expression of ExsA [85], which, in turn, is heterogeneously expressed, leading to variation in T3SS gene expression upon activation. In the absence of activation signals, this heterogeneous regulation through ExsA generates a primed T3SS subpopulation, ready to rapidly induce effector gene expression in response to signals [86]. Within the same genus, the plant pathogen *Pseudomonas syringae* demonstrates bistability in its T3SS expression [25]. This bistability is governed by different feedback loops within the hrp regulon, which encodes the T3SS, regulators, and effectors. The main transcriptional activator, HrpL, controls the hrp regulon, with HrpA involved in a positive feedback loop, HrpV in a negative feedback loop, and HrpG acting as an anti-negative feedback loop [25,87]. These regulatory cascades involving multiple feedback processes provide an ideal environment for the emergence of bistability, which can be activated by signals or stochastic events.

## 4. Flagellum Nanomachine

### 4.1. Flagellum Structure and Functions

The flagellum, a ubiquitous bacterial nanomachine, facilitates motility and navigation through diverse environments. Evolutionarily related to the injectisome [88], its structure is well-conserved and consists of a basal body anchored in the membrane. This basal body is composed of multiple rings (OM-, IM-, and C rings) and a rod that houses the type III secretory apparatus. (Figure 3a) [15,71,89]. The flagellar hook connects the basal body to the filament made of flagellin protein subunits. It acts as a propeller when torque is applied by a proton-motive force-powered motor located at the inner membrane [15,71].

Bacterial motility, primarily achieved through swimming and swarming mechanisms, enables bacteria to colonize new environments and evade threats [18]. Swimming involves propelling through liquid environments [90] while swarming on solid surfaces involves upregulated flagella and surfactant production for surface gliding [90]. Chemotaxis, regulated with flagellar structure genes [91,92], allows bacteria to navigate gradients of nutrients and signaling molecules to locate optimal niches for survival [18,93]. The flagellum plays a significant role in biofilm formation by promoting auto-aggregation and the initial attachment to surfaces. Despite its repression, it also serves as a structural element of the biofilm matrix [18]. This multifunctionality contributes to the adaptability and success of bacteria across diverse ecological niches, reasons why this nanomachine is highly conserved in many bacterial species.

### 4.2. Flagellar Regulatory Pathway

The flagellar regulatory network, extensively studied in organisms such as *Salmonella enterica* and *Escherichia coli* [91], is intricately linked to the assembly of the flagellar structure. Genes in this pathway are divided into three temporal classes: early, middle, and late (Figure 3b). The master flagellar operon, *flhDC*, contains early genes encoding FlhC and FlhD proteins, which positively activate the transcription of middle genes essential for the basal body and hook assembly. FlgM and FliA (σ28) are also expressed in that class. FlgM binds to FliA to prevent the expression of late genes until hook-basal body assembly is complete. Once assembled, FlgM is secreted, freeing FliA to initiate transcription of late genes encoding filament proteins. This regulatory mechanism ensures filament synthesis only occurs when the base structure is ready for assembly [91].

The assembly pathway of the flagellum is largely regulated by environmental cues and cellular states that act on the class I promoter region, controlling the expression of the early *flhDC* genes [91]. These signals include cAMP-CRP, temperature, heat shock proteins, osmotic stresses, and cell-cycle events [91]. Additionally, in some species like *Pseudomonas aeruginosa*, cyclic-di-GMP, binding to the major regulator FleQ [94], inhibits flagella formation upon host cell contact [28]. Quorum sensing also plays a role in flagellar regulation in various bacteria, such as Legionella pneumophila. The detection of signaling molecules in *Legionella* triggers a cascade of events that leads to the expression of flagellar genes by relieving repression mediated by CsrA [95].

### 4.3. Molecular Bases of the Flagellum Bistability

Flagellar bistability in bacteria involves the coexistence of two distinct subpopulations: one with flagella and motility and one without. Different regulatory pathways govern this phenomenon across bacterial species. In Escherichia coli, flagellar regulation is driven by a pulsatile mechanism. Continuous expression of *flhDC* leads to spontaneous switching between active and inactive transcription of downstream flagellar genes in a rhythmic pattern [27]. The specific mechanism underlying this pulsatile expression is still not fully understood. However, it does not require specific transcriptional or translational regulation of the flagellar master regulator, FlhDC, but instead appears to be essentially governed by an autonomous post-translational circuit due to transcriptional “noise”. The authors hypothesized that *E. coli* has evolved to operate its flagellar synthesis network in a “pulsatile” mode. This allows each cell to intermittently sample various flagellar phenotypes rather than committing a subpopulation to fixed flagellar expression over extended periods [27]. In *Salmonella enterica*, different pathways regulate flagellar bistability. The EAL-like protein RflP inhibits flagellum expression [96], resulting in bistability when RflP represses flagellin synthesis, leading to phenotypic heterogeneity in the Salmonella-containing vacuole [29]. The expression of RflP is triggered by cell envelope stress, such as low pH and membrane modifications, and it regulates the degradation of the FlhDC complex via ClpXP proteases [29]. However, the exact reasons behind the bistable expression of rflP remain unclear.

In *Pseudomonas aeruginosa*, it has been demonstrated that upon encountering surfaces, there is a rapid increase in the concentration of cyclic-di-GMP [97,98]. This secondary messenger influences various cellular processes, notably regulating the transition between sessile and planktonic lifestyles [99]. This elevation in cyclic-di-GMP levels promotes surface adherence and induces virulence through the FimW receptor [98,100]. Subsequently, the attached bacteria undergo asymmetric division, resulting in the generation of two distinct cell types: one piliated, surface-anchored cell and one flagellated, motile spreader. This differentiation is controlled by a phosphodiesterase localized at the flagellated pole, which maintains low levels of cyclic-di-GMP in the future motile cell side, allowing uninhibited flagellar motility [28,98,100]. Similar mechanisms have been identified in non-pathogenic bacteria such as *Caulobacter crescentus* and the opportunistic pathogen *Shewanella* spp. [28].

In *Clostridium difficile*, a fascinating mechanism involving heterogeneity in motility and toxin production has been described, driven by DNA recombination orchestrated by the recombinase RecV [101]. This process involves the reversible inversion of a flagellar switch sequence located between the promoter and the flagellar biosynthesis operon *flgB* by RecV, resulting in toggling between the ON/OFF expression of the flagellum. When the switch sequence is in the OFF direction, the transcription termination factor Rho binds to the mRNA, halting transcription and preventing flagellar operon expression [31]. However, this phase variation involves a reversible genetic change. The question arises as to whether this can be considered phenotypic heterogeneity.

Some other mechanisms are found. In *Bacillus subtilis*, stochastic expression of the transcription factor SigD at the end of a long operon triggers its own expression, acting like a positive feedback loop and that of basal body and filament operons of the flagellum once a threshold is reached [26,102,103], leading to the emergence of motile and non-motile subpopulations. Furthermore, in *Legionella pneumophila*, intraspecies quorum sensing appears to drive the appearance of a transmissive flagellate subpopulation at the *Legionella*-containing vacuole periphery [30]. However, this mechanism does not fully explain bistability, as heterogeneity persists even in quorum sensing-deficient strains.

## 5. Nanomachines Bistability and Pathogen Community Behavior

Phenotypic heterogeneity of bacteria can have a significant link with their social behavior, particularly in contexts such as biofilm formation or cooperation/competition within a bacterial population. The phenotypic heterogeneity of bacteria enables functional diversity that promotes survival and proliferation in diverse and changing environments. This diversity is essential for the social behavior of bacteria, allowing them to cooperate, compete, or adapt effectively to fluctuating environmental conditions. With provided evidence that nanomachines are subject to bistability in pathogenic bacteria, the most important question remains: How does bistability shape and impact pathogens’ life cycle? Does it provide a gain of function at the population levels? And if yes, which functions? Does it have an impact on population behavior?

### 5.1. Persistence against Antibiotics

At the population level, phenotypic heterogeneity provides a crucial advantage, enabling certain subpopulations to endure fluctuating environmental conditions through a strategy known as bet-hedging [2,8,104,105]. By diversifying behavior, growth rates, and other traits, microbial populations increase their resilience to changing environmental pressures. This adaptive strategy involves hedging bets across various phenotypic possibilities, ensuring that at least some individuals are more adapted to current or future conditions. In pathogenic contexts, this strategy enhances the likelihood of survival and success within-host environments, where pathogens encounter diverse and dynamic conditions, including fluctuations in immune responses, nutrient availability, and antibiotic exposure. By maintaining a subset of the population with diverse traits, pathogens can evade host immune responses, persist within the host over extended periods, and establish infections successfully [3].

Nanomachines, such as the injectisome, contribute significantly to pathogen fitness at the population level during infection conditions [16,18,106]. While the expression of the injectisome facilitates the establishment of host infections, it comes at a fitness cost for the overall machinery expression. For instance, in *Salmonella*, subpopulations expressing the Type III Secretion System (T3SS) exhibit significantly slower growth rates [107]. This reduced growth rate can confer tolerance to environmental stressors, including increased antibiotic resistance (Figure 4a) [68,108,109]. The phenomenon of nanomachine bistability raises questions regarding its role in the formation of persister cells, which enter a dormant state, protecting them from environmental stresses and the lethal effects of antimicrobial compounds. Bacterial persisters play a significant role in the persistence of chronic infections and contribute to therapeutic failures, as they can later resume growth and reinstate infections [109,110].

Similarly, still in *Salmonella enterica*, flagellum assembly, and rotation consume proton motive force, resulting in decreased efflux pump activity [111]. This trade-off between motility and efflux activity induces greater antibiotic resistance in flagellum-non-expressing bacteria under antibiotic stress compared to express ones.

### 5.2. Cooperative Colonization

Division of labor is characterized by cooperation between specialized subpopulations to enhance the overall efficiency and functionality of the population [2,3,112]. Unlike bet-hedging, which relies on phenotypic diversity to cope with unpredictability, division of labor optimizes group fitness and adaptability by assigning different tasks to specialized members [2,3,112]. In infections, pathogens utilize this principle to reinforce collective efficiency across various infection stages. Within a host, distinct pathogenic cell subsets may specialize in replication, colonization, or virulence tasks [100,113].

In *Pseudomonas aeruginosa*, the injectisome expression facilitates antiphagocytic activities via the ExoS effector [114]. However, this expression is also essential for intracellular survival and growth. The non-expressing subpopulation enables host cell invasion [115]. Subsequent to invasion, phenotypic variations emerge, with T3SS expressed within bronchial epithelial cells, enabling bacteria to exit the vacuole and colonize the host cell cytosol and tissues. This cooperative strategy establishes invasion in multiple niches, with replicating cells at the cell surface, invasive cells in epithelial cell vacuoles for persistence, and cytosol-associated cells promoting infection dissemination [116]. This approach favors both short- and long-term population survival and may potentially lead to therapeutic failures.

The bistability of the flagellum plays a pivotal role in facilitating cooperative processes within bacterial communities. This dual-state capability allows bacteria to switch between flagellated and non-flagellated states depending on environmental conditions or signals from neighboring cells. Such flexibility enables coordinated movement, collective behaviors like biofilm formation, and efficient utilization of resources. By harnessing flagellum bistability, bacteria can optimize their survival strategies and adaptability in diverse and challenging environments. In *Pseudomonas aeruginosa*, asymmetrical division results in a mother cell anchored to epithelial cells and a spreading flagellated daughter [100]. This allows attached “striker” cells to initiate local host attacks, while planktonic “spreader” cells disperse infection to other sites (Figure 4b). Upon touching another epithelial surface, spreaders attach via type IV pili and initiate asymmetric division to colonize the entire epithelial lining [100]. A similar strategy is observed in *Legionella pneumophila* [30]. In contrast, flagellated bacteria facilitate host cell exit in late infection, enabling the spread and colonization of new hosts.

### 5.3. Cooperative Virulence

The cooperation between injectisome and flagellum bistable populations has been demonstrated in Salmonella enterica [117,118]. Both nanomachines are highly conserved across bacterial species and function as a pathogen-associated molecular pattern (PAMP), triggering host immune responses during infection [19,20]. The subpopulation non-expressing injectisome thrives in the gut lumen, focusing on replication, while the expressing subpopulation infiltrates tissues, eliciting immune responses (Figure 4c). For the flagellum non-expressing subpopulation, it enables the evasion of eukaryotic defense pathways at the host level and even within macrophage intracellular infections by preventing the caspase-1 inflammatory response [118]. Conversely, flagellum-expressing bacteria trigger an inflammatory cascade, leading to the pyroptosis of macrophages [119]. The inflammation leads to the death of the expressing subpopulation. Nevertheless, it eliminates competing bacteria, thereby outcompeting the natural host microbiome. Despite the fitness cost and mortality of the invading bacteria, the expressing subpopulation benefits the counterpart residing in the lumen [117].

### 5.4. Evolution

Pathogens infect hosts through coordinated actions, such as secreting growth-promoting compounds or virulence factors or triggering host responses that aid in colonization. However, these behaviors are susceptible to the emergence of mutants that exploit the benefits of collective action without contributing to it, which complicates their evolutionary stability. In their study on *S. Typhimurium*, Diard et al. demonstrated the critical role of a phenotypically avirulent subpopulation, genetically identical, in maintaining the evolutionary stability of virulence [113]. They observed the intra-host evolution that gives rise to genetically avirulent, fast-growing mutants. These mutants act as defectors, leveraging inflammation without enhancing it. In infection experiments with wild-type *S. Typhimurium*, defectors slowly increase in frequency. These findings highlight that *S. Typhimurium*’s manipulation of the host represents a cooperative trait vulnerable to exploitation by avirulent defectors. The presence of a phenotypically avirulent subpopulation, growing at a rate similar to defectors, slows this process, thereby bolstering the evolutionary stability of virulence. This underscores the pivotal role of bistable virulence gene expression in stabilizing cooperative virulence and suggests novel strategies for controlling pathogens.

## 6. Conclusions and Outlook

Bacterial populations display phenotypic heterogeneity, an evolutionarily conserved phenomenon that facilitates complex community behaviors despite minimal genetic diversity and potentially small population sizes of just a few dozen individuals. This phenotypic diversity is especially critical during infection, enabling monoclonal pathogen populations to persist [120], thrive [116], and adapt [121] within challenging and variable environmental conditions prior to and during antibiotic treatment.

The nanomachines, particularly the injectisome, and flagellum, exhibit bistable expression, a phenomenon crucial for bacterial adaptation and fitness across diverse environments. Bistability involves multiple regulatory mechanisms that vary among microbial species, enabling two distinct survival strategies within a population: bet-hedging for stress persistence and division of labor for cooperative behavior [104,112]. In pathogens, this bistability influences disease progression, chronicity, and treatment outcomes by promoting tissue colonization and the emergence of persister cells.

While the flagellum and injectisome are well-studied for bistable expression, other nanomachines like the pilus and type 6 secretion system (T6SS) also demonstrate bistability in bacteria such as *Pseudomonas aeruginosa*. These systems, associated with the sessile lifestyle, are inversely regulated with the flagellum via c-di-GMP signaling pathways [122,123]. Additionally, pilus gene expression in *Vibrio cholerae* [124] and epigenetic regulation in *Streptococcus pneumoniae* [125], along with T6SS expression in *Acinetobacter baumannii*, exhibit intriguing bistability mechanisms [126].

Despite its significance in pathogenesis, nanomachine bistability remains understudied, with only a fraction of bacteria demonstrating described bistable mechanisms. Unraveling these complex mechanisms holds implications for various research fields, yet many questions persist. From an ecological standpoint, understanding the contribution of bistability to bacterial life cycles in environmental habitats and its impact on ecosystem dynamics and interspecies interactions remains a challenge. At the single-cell level, exploring the extent of heterogeneous gene expression and its broader phenotypic implications is essential. In the context of pathogenesis, there is limited information concerning the role of host factors in driving phenotypic expression and the host response to bacterial bistability [127,128,129,130], elucidating it could lead to novel therapeutic strategies.

## Figures and Tables

**Figure 1 microorganisms-12-01930-f001:**
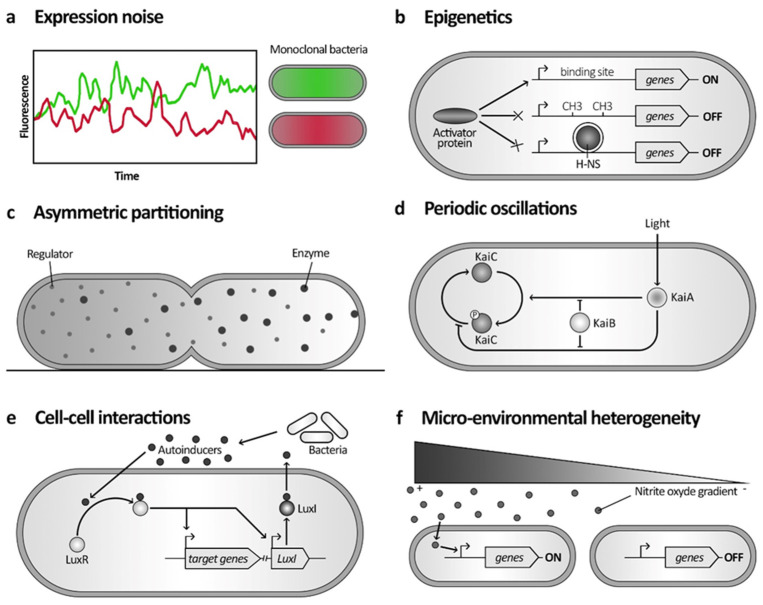
Molecular mechanisms governing phenotypic heterogeneity. Schematic representation of molecular pathways involved in different phenotypic heterogeneity mechanisms. (**a**) The expression of two genes may become uncorrelated in individual cells due to intrinsic expression noise, resulting in a population where some cells express higher levels of one fluorescent protein (green) compared to the other (red). (**b**) Epigenetics can prevent gene expression by DNA methylation patterns or histone-like nucleoid-structuring proteins (H-NS) in the promoter region. It modifies chromosome conformation and accessibility for the activator proteins, resulting in the switching off of downstream genes. (**c**) Asymmetrical division in bacteria can contribute to phenotypic heterogeneity through the stochastic segregation of cytoplasmic molecules. Different regulators and enzymes, for instance, can affect the gene expression profile of each cell after division. (**d**) Periodic oscillations in *Synechococcus elongatus* form a circadian clock that is influenced by light. This process revolves around the phosphorylation of KaiC by KaiA, which can be either inhibited or enhanced depending on light conditions. KaiB acts to counteract both of these effects. (**e**) Quorum sensing is a cell–cell interaction via diffusible molecules: autoinducers (AI). In *Vibrio fisheri*, LuxI is N-acyl homoserine lactone synthetase, which generates and secretes AI. Other cells bind AI on LuxR, influencing gene expression and promoting the production of more AI. (**f**) Microenvironmental heterogeneity of nitric oxide (NO) levels at infection sites leads to varying gene activations within the Yersinia pseudotuberculosis population in response to NO molecules.

**Figure 2 microorganisms-12-01930-f002:**
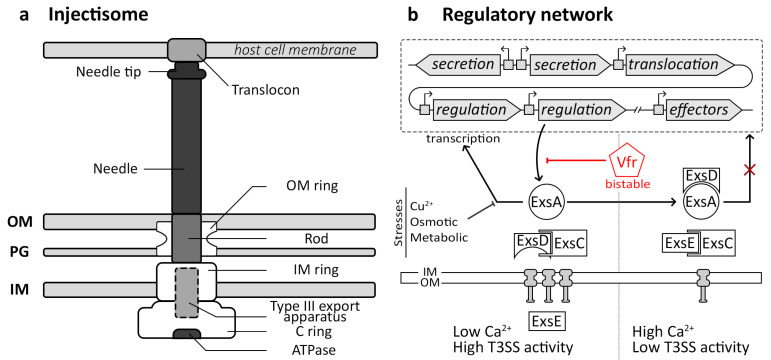
Structure and regulation of the injectisome. Schematic comparison of structures and regulations of the injectisome and the flagellum. (**a**) Schematic overview of the injectisome structure. The outer membrane (OM) ring, the inner membrane (IM) ring and the cytosolic C ring are principal components stabilized by the membranes and the peptidoglycan (PG). (**b**) Simplified genetic organization and regulation of the *Pseudomonas aeruginosa* injectisome (T3SS). ExsA-binding sites (square) are located upstream of all T3SS genes. In high Ca^2+^ conditions, ExsE binds ExsC, and ExsD binds ExsA, preventing the regulon transcription. In low Ca^2+^ and high secretion activity, ExsE is secreted, freeing ExsC that binds ExsD and ExsA that can bind to the gene’s promoter region and allow the injectisome gene to be expressed.

**Figure 3 microorganisms-12-01930-f003:**
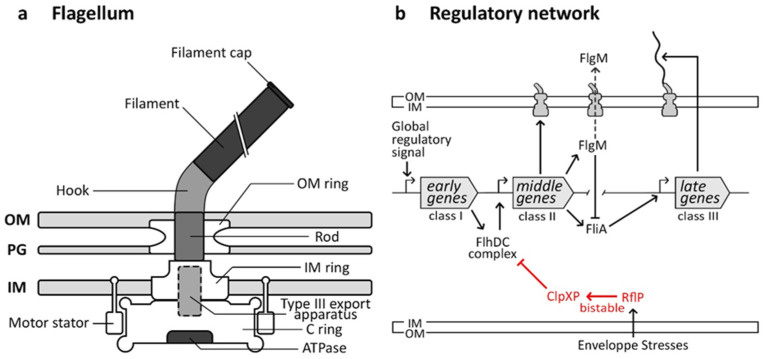
Structure and regulation of the flagellum. Schematic comparison of structures and regulations of the injectisome and the flagellum. (**a**) Schematic overview of the flagellum structure. (**b**) Simplified regulation of *Salmonella* and *Escherichia coli* flagellum. Global signals induced the expression of class I genes encoding the FlhDC protein complex. It promotes the expression of class II genes encoding basal body structures and regulators FlgM and FliA. The latter is an activator of the class III genes encoding for the flagellar filament. FlgM binds FliA and prevents the class III transcription. Once the hook and basal body are assembled, FlgM is secreted, freeing FliA.

**Figure 4 microorganisms-12-01930-f004:**
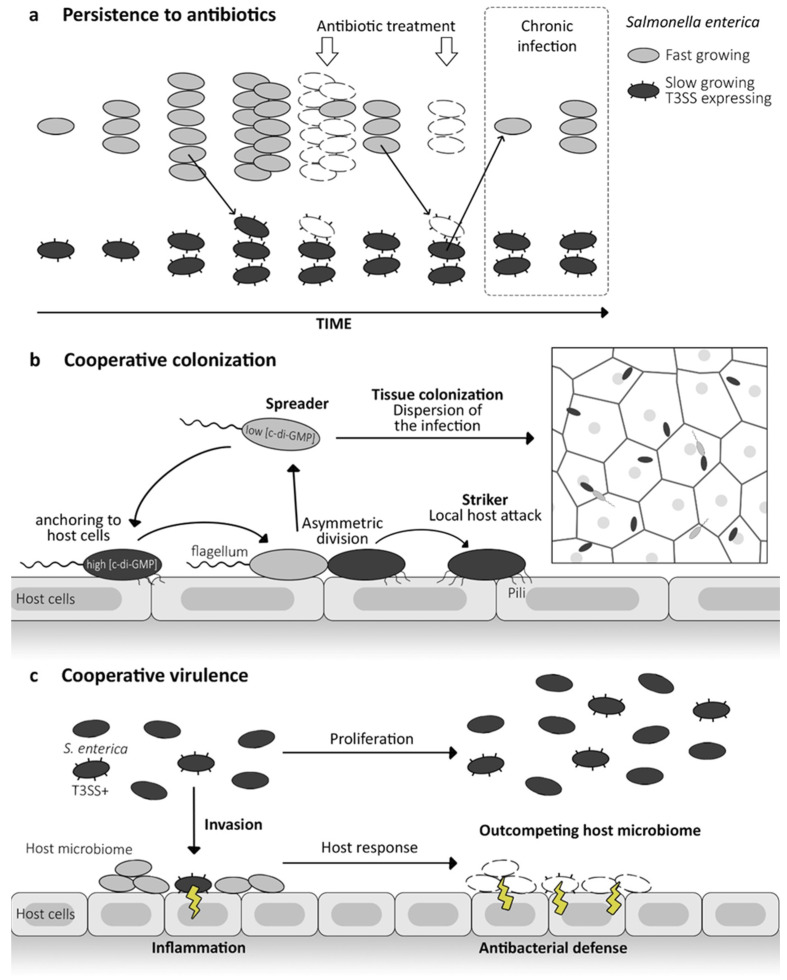
Bistability in pathogens’ life cycle: bet-hedging and division of labour. (**a**) Bet-hedging in *Salmonella enterica* infections. Expression of the injectisome (T3SS; type III secretion system) leads to slow growth of bacterial cells. This slower metabolism makes bacteria less sensitive to antibiotics and can result in the formation of persisters; the T3SS-expressing subpopulation is more likely to survive the stress. These resistant, slow-growing bacteria can then switch back or divide into fast-growth cells and cause chronic infections. (**b**) Cooperative colonization in *Pseudomonas aeruginosa* infections. When a flagellated cell encounters host cells, the cyclic-di-GMP level increases, allowing for the expression of pili. This is followed by an asymmetric division, with the c-di-GMP being polarly located and having a lower concentration at the flagellated pole. The daughter cell with a lower c-di-GMP level (light grey) becomes a spreader and is involved in dispersing the infection, while the high c-di-GMP (dark grey) cell remains anchored to the host and provides a local attack. (**c**) Cooperative virulence in *Salmonella enterica* gut infection.

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
