# Peer review of "Unravelling the Roles of Bacterial Nanomachines Bistability in Pathogens’ Life Cycle"

_microorganisms, 2024, doi:10.3390/microorganisms12091930_

Round 1

Reviewer 1 Report

Comments and Suggestions for Authors

Review by Romain Gory  et al. “Uncovering the role of bistability of bacterial nanomachines in the life cycle of pathogens” is devoted to the description of two categories of bacterial nanomachines, type 3 secretion system and the flagellum, and their role in underscoring their dynamic and complex regulation as well as their contribution to shaping the bacterial community behavior during the infection process

An interesting review that summarizes the data obtained on bacterial nanomachines such as injectosomes and bacterial flagella. The authors covered the problem well and cited a large number of publications.

 The manuscript can be accepted for publication after a small revision. There are a few typos.

Line 232 “Injectisome structure and fonctions” – there is a typo (functions)

Line 287, 289  Ca2+ - it should be superscript

Line 397 “E. coli” all Latin names should be in italics.

Line 436 Please, explain “The PH” before abbreviation

Author Response

dear Reviewer 1: Thank you for reading and improving our article. We have taken all typos into account and corrected them.

Reviewer 2 Report

Comments and Suggestions for Authors

Dear authors, congratulations on the excellent work. Below I leave some considerations:

The text of the introduction makes broad generalizations without providing specific details. For example, the statement "for decades, it has been conventionally believed that microbial community behavior is fueled by genetic diversity" is vague and lacks specific, historical context. Terms such as "often" and "potential" are used ambiguously, leaving the reader with doubts about the frequency and relevance of the phenomena described. The passage lacks logical coherence and flow. The transition between topics such as "genetic diversity", "phenotypic heterogeneity" and "bacterial nanomachines" is abrupt and disorganized. The structure of the text is confusing, with excessively long and complicated sentences that make it difficult for the reader to understand. Lack of clarity can be seen in phrases like "stable states under the same set of conditions", which are unnecessarily complicated. Complex concepts such as "bistability" and "phenotypic heterogeneity" are introduced without adequate explanations or concrete examples, leaving the less specialized reader lost.

Author Response

Reviewer 2
Comments and Suggestions for Authors
Dear authors, congratulations on the excellent work. Below I leave some considerations:
The text of the introduction makes broad generalizations without providing specific details. For example,
the statement "for decades, it has been conventionally believed that microbial community behavior is
fueled by genetic diversity" is vague and lacks specific, historical context. Terms such as "often" and
"potential" are used ambiguously, leaving the reader with doubts about the frequency and relevance of
the phenomena described.
The passage lacks logical coherence and flow. The transition between topics such as "genetic
diversity", "phenotypic heterogeneity" and "bacterial nanomachines" is abrupt and
disorganized. The structure of the text is confusing, with excessively long and complicated sentences
that make it difficult for the reader to understand. Lack of clarity can be seen in phrases like "stable
states under the same set of conditions", which are unnecessarily complicated.
Complex concepts such as "bistability" and "phenotypic heterogeneity" are introduced without adequate
explanations or concrete examples, leaving the less specialized reader lost.
Answer :
We thank the reviewer for being positive on the manuscript as well as for the clear
suggestions in order to improve the paper. We have taken into account comments on
ambiguous words in the text and modified or deleted them to make the document clearer.
We try to better explain the relationship between as "genetic diversity", "phenotypic
heterogeneity" and "bacterial nanomachines"
The reviewer talk about: complex concepts such as "bistability" and "phenotypic
heterogeneity" that are introduced without adequate explanations or concrete examples,
leaving the less specialized reader lost. To answer this remark, in the introduction we
propose 3 references about examples of bistability (references 9, 10 and 11: lines 43 to 47.
Regarding “phenotypic heterogeneity”, we have several references about examples (8, 11).
To answer to remark of the reviewer we simplify some complex sentences as recommended.

Reviewer 3 Report

Comments and Suggestions for Authors

This is an interesting review that explores the mechanisms behind bistability of two bacterial nanomachines: the flagellum and the injectosomes.  It contains a thorough literature revision and fits well with the journal's scope.

I  offer the following points to the authors for manuscript strength:

- It would be relevant to include in the listed mechanisms the role of mutations on bistability. Reversion, suppression, and more relevant adaptative mutagenesis are factors that can play a role in this phenomenon. Moreover, mutations that affect the pole of ribonucleotides and the translation-associated repair mechanisms may be relevant.

- It would be relevant to include some information on how bistability may affect the bacteria-immunity interaction, in particular with cellular effectors.

Author Response

Reviewer 3
Comments and Suggestions for Authors
This is an interesting review that explores the mechanisms behind bistability of two bacterial
nanomachines: the flagellum and the injectosomes. It contains a thorough literature revision and fits
well with the journal's scope.
We thank the reviewer for its positive feedback on the review and his/her suggestions to improve the
manuscript.
I offer the following points to the authors for manuscript strength:
- It would be relevant to include in the listed mechanisms the role of mutations on bistability. Reversion,
suppression, and more relevant adaptative mutagenesis are factors that can play a role in this
phenomenon. Moreover, mutations that affect the pole of ribonucleotides and the translation-associated
repair mechanisms may be relevant.
In this review, we focus on the phenomenon of phenotypic heterogeneity. The bacterial phenotypic
heterogeneity is the ability for a clonal population of bacteria to reversibly produce bacterial individuals
with distinct phenotypes so the population gain new functionalities. The bistability is one expression of
such coexistence of diverse phenotypes. The term bacterial phenotypic heterogeneity underlies that
there is no edition of the bacterial genomes (eg. mutations). Contrary to mutagenesis, the bacterial
phenotypic heterogeneity is a way to reversibly diversify the bacterial physiology in order to deal with
highly fluctuant environment (short-term / reversible adaptation). Thus, in order to keep the scope of the
review we do not discuss in-depth the interconnection between mutagenesis, synthetic biology and
bistability. This told, in the section 5 entitled “Nanomachines bistability and Pathogen community
behavior” subheading 5.4 “Evolution” p 14 (L582-597) we briefly mention the interconnection between
bacterial bistability (as expression of bacterial phenotypic heterogeneity) and bacterial mutagenesis. We
notably highlight the critical role of a the bistability in maintaining the evolutionary stability of virulence
(work by Diard et al., 2013).
- It would be relevant to include some information on how bistability may affect the bacteria-immunity
interaction, in particular with cellular effectors.
Few studies have investigated if bistability of bacterial nanomachines impacts the bacteria-immunity
interaction. Our group has recently published an Opinion connecting such bistability, innate immunity
detection and bacterial persister formation. It remains yet speculative (Dadole I., et al., 2024;
DOI: 10.1016/j.tim.2024.02.009). In 2015, Roi Avraham and colleagues combined single-cell RNA-seq
with fluorescent markers to probe the responses of individual macrophages to invading Salmonella.
They found a possible bistability in the bacterial LPS modifications driving variable host type I IFN
responses (DOI: 10.1016/j.cell.2015.08.027). This work does demonstrate a causative link between host
and bacterial variability but it remains out of the scope of the review that focuses on the T3SS and the
flagellum.